# Hepatitis Risk in Diabetes Compared to Non-Diabetes and Relevant Factors: A Cross-Sectional Study with National Health and Nutrition Examination Survey (NHANES), 2013–2018

**DOI:** 10.3390/ijerph20064962

**Published:** 2023-03-11

**Authors:** Ja-Young Han, Jae-Hee Kwon, Sun-Hwa Kim, Heeyoung Lee

**Affiliations:** 1Department of Clinical Medicinal Sciences, Konyang University, Nonsan 32992, Republic of Korea; 2College of Pharmacy, Graduate School of Pharmaceutical Sciences, Ewha Womans University, Seoul 03760, Republic of Korea

**Keywords:** diabetes mellitus, hepatitis B, hepatitis C, illicit drug, National Health and Nutrition Examination Survey (NHANES), poverty

## Abstract

This study aimed to identify the development of hepatitis B or C infection in diabetes patients compared to those without and to elucidate factors associated with the prevalence of hepatitis B or C infection in diabetes. We conducted a cross-sectional study using data from the National Health and Nutrition Examination Survey (NHANES) 2013–2018. As evaluation factors, we included variables such as age, race, illicit drug use, and poverty. The diabetic group had a significantly higher prevalence of hepatitis B or C infection than the non-diabetic group (odds ratio (OR) = 1.73; 95% confidence interval (CI), 1.36–2.21, *p* < 0.01). In multivariate Cox regression, non-poverty and non-illicit drug use were lower risk factors contributing to hepatitis development in diabetes (hazard ratio (HR) = 0.50; 95% CI, 0.32–0.79, *p* < 0.01, and HR = 0.05; 95% CI, 0.03–0.08, *p* < 0.01, respectively). Logistic regression also showed that these factors were significant contributors to hepatitis development in the diabetic group (*p* < 0.01). In patients with diabetes, the development of hepatitis was higher than that in those without, and hepatitis development was influenced by poverty and illicit drug use. This may provide supporting evidence of response strategies for diabetes to care for hepatitis development in advance.

## 1. Introduction

Diabetes and hepatitis are among the most prevalent diseases worldwide [1]. The concomitant existence of diabetes and hepatitis potentially leads to a life-threatening status, increasing mortality by approximately 17% among diabetic patients without comparing to other patients, such as non-virus-infected diabetic patients [2]. With a considerably high prevalence, 865 outbreaks of patients infected with hepatitis B virus (HBV) through population-based surveillance for infectious diseases at eight Emerging Infections Program (EIP) sites, which are part of a network of 10 state health departments in the United States [3], have been reported among adults diagnosed with diabetes [4]. Regarding the increased prevalence of HBV in patients with diabetes and severe clinical outcomes, the Advisory Committee on Immunization Practices (ACIP) of the Centers for Disease Control and Prevention (CDC) recommends appropriate methods for preventing HBV infection in patients with diabetes based on the risk of an inadequate immune response [5]. In addition, for hepatitis C virus (HCV) infection, one of the major sources of morbidity and mortality in people, the CDC indicated that injection drug use (IDU) is a primary risk factor for infection [6]. Considering IDU use in diabetic patients, according to the guidelines for diabetes, diabetic patients are given insulin as the primary treatment and use self-monitoring of blood glucose (SMBG) to manage their blood glucose, so frequent exposure to needles increases the risk of HCV transmission [7,8]. Regarding the frequent exposure to IDU along with needles through multiple blood sampling and unsafe injection practices, patients with diabetes are more vulnerable to HCV infection than those without diabetes [9]. A previous cross-sectional study which analyzed the prevalence of HCV among diabetic patients reported that insulin users were 3.2 times more likely to have HCV infection than non-insulin users [10].

Furthermore, hepatitis and diabetes seem to be negatively associated with the development of each other. Schillie et al. [11] reported a higher prevalence rate of HBV infection among persons with diabetes than those without diabetes (odds ratio (OR) = 1.60; 95% confidence interval (CI), 1.30–1.90, *p* < 0.05). However, there is still controversy regarding the association between the development of diabetes and hepatic viral infection, as shown in a study in a tertiary hospital reporting low development of hepatitis C viral seropositivity among patients with type 2 diabetes mellitus [12]. To provide more substantial evidence explaining the association between diabetes and hepatitis, including HBV or HCV, large-scale epidemiological studies evaluating national data, such as the National Health and Nutrition Examination Survey (NHANES), are needed.

Currently, a limited number of studies have analyzed large-scale data, such as the NHANES, which might provide a clear explanation of the association between diabetes and hepatitis. Thus, to provide sufficient evidence to evaluate the impact of diabetes on HBV or HCV development compared to the population without diabetes, the current study examined the association between hepatitis infection and diabetes and aimed to characterize the factors related to the prevalence of hepatitis in the diabetic population using the NHANES database.

## 2. Materials and Methods

### 2.1. Study Design

We analyzed the 2013–2018 NHANES database. The NHANES is a cross-sectional monitoring program designed to assess the health and nutritional status of adults and children in the United States [13]. Multiple datasets were collected in this survey, including demographics, dietary, questionnaire, physical examinations, and laboratory testing of biologic samples in the U.S. populations [13]. The NHANES data have been collected in 2-year cycles without a break between cycles since 1999 [14]. The methodology for all databases is described on the NHANES website [14]. Since the participants of the NHANES are de-identified and assigned unique sequence numbers, details of the respondents and duplicate participation of people are unknown [15]. The NHANES is administered using a stratified multistage clustered probability sampling strategy to provide a nationally representative sample [15].

### 2.2. Definition of Diabetes and Hepatitis 

Information on diabetes, HBV, and HCV were retrieved from the NHANES. Diabetes was defined according to the following criteria: HbA1c ≥ 6.5%, FPG ≥ 126 mg/dL, or participants being informed that they had diabetes by their doctor or other health professionals [16]. Both patients with type 1 and type 2 diabetes were included in our study. HBV was defined as positive for hepatitis B surface antigen (HBsAg) or if participants answered yes to the question, “Has a doctor or other health professional ever told you that you have hepatitis B?” [17]. HCV was defined as a positive for Hepatitis C virus ribonucleic acid (HCV-RNA) or if participants replied yes to the question “Has a doctor or other health professional ever told you that you have Hepatitis C?” [18]. HBV-infected and HCV-infected participants were assigned to the corresponding hepatitis B and C groups.

### 2.3. Covariates 

We selected significant variables through logistic regression for variable selection to be included in the analysis dataset [19]. In our analysis, the confounding variables included age, sex, race, body mass index (BMI), the ratio of family income to poverty, and the use of illicit drugs. Demographic covariates included age (years) [20], sex (male or female), race (Mexican-American, other Hispanic, non-Hispanic White, non-Hispanic Black, non-Hispanic Asian, or other race) [21], and family income to poverty ratio (continuous from 0 to 4.99) [18]. We categorized age into four groups as follows: 0–19, 20–44, 45–64, and ≥65 years [20]. The family income to poverty ratio was divided into < or ≥1 [19]. A ratio < 1 indicated poverty, while a ratio ≥ 1 indicated without poverty [18]. BMI was collected from the laboratory dataset from the NHANES database and grouped as underweight (<18.5 kg/m²), normal weight (18.5~24.9 kg/m²), and overweight (≥25 kg/m²) [22]. Questionnaire covariates included the use of illicit drugs (yes or no) [23].

### 2.4. Statistical Analysis

Data are presented as frequency and percentage for categorical variables. Box plots consisted of quartiles for the prevalence over 2-year cycles of patients infected with hepatitis regarding diabetes status [24]. The OR with 95% CI was calculated using univariate and multivariate logistic regression models to elucidate the factors affecting the development of hepatitis in patients with diabetes. Multivariate logistic regression analysis was performed after adjusting for covariates of race, poverty, and illicit drug use. The Cox proportional hazards model determined the hazard rate to be roughly constant at all time points, so our study used this statistical method [25,26]. The Cox proportional hazards model was used to calculate the hazard ratio (HR) and 95% CI of the risk associated with hepatitis development in patients with diabetes. The univariate Cox proportional hazards model was used for all variables, and a multivariate model was utilized to adjust the race, poverty, and use of illicit drugs. All statistical analyses were performed using R software (version 4.2.1), with a *p*-value < 0.05 as statistically significant.

## 3. Results

### 3.1. General Characteristics of Participants

This study included 29,400 participants from the NHANES (2013–2018) database. Overall, 1160 participants with missing diabetes data were excluded. Then, 15,305 participants with missing data on illicit drug use, 28 participants with missing data on HBV, 3 participants with missing data on HCV, 1215 participants with missing data on the ratio of family income to poverty, and 88 participants with missing data on BMI were excluded in the present analysis. The additional analysis to clarify the variance between excluded data, including data related to illegal drug use, showed that the number of participants differed among age groups (Appendix A). However, the distributions of excluded data for non-responding questions related to illegal drugs were not significantly different while the number of participants distributed among age groups differed. Ultimately, 11,601 participants were included in the final analysis (Figure 1 and Table 1). Of the 11,601 participants, 1709 were diagnosed with diabetes, whereas 9892 were classified in the non-diabetic group. The prevalence of HBV, HCV, and hepatitis B or C with diabetes was 2.3%, 3.2%, and 5.1%, respectively, whereas the prevalence in the non-diabetic group was 1.3%, 1.9%, and 3.0%, respectively (Table 2). 

### 3.2. The Prevalence of Hepatitis According to Diabetes Status

The prevalence of HBV in participants without diabetes was 1.3%, whereas that in participants with diabetes was 2.3%, which indicated HBV development was 1.7721 odds more than in patients with diabetes, with a significant difference (OR = 1.77; 95% CI, 1.24–2.53, *p* < 0.05). The prevalence of HCV was 3.2% among patients with diabetes and 1.9% among those without diabetes. The development of HCV increased in patients with diabetes by 1.6751 odds compared to patients without diabetes, with a significant difference (OR = 1.68; 95% CI, 1.23–2.28, *p* < 0.05). Moreover, the prevalence of 5.1% for hepatitis B or C in patients with diabetes was significantly 1.7298 odds higher than that reported among non-diabetic subjects (OR = 1.73; 95% CI, 1.36–2.21, *p* < 0.0001) (Figure 2, Table 2 and Appendix A). Furthermore, the prevalence of hepatitis infection was not significantly different based on the FPG or HbA1c levels (Appendix A and Appendix A). 

### 3.3. Characteristics of Hepatitis Development from 2013 to 2018

Analysis of the characteristics of hepatitis prevalence according to diabetes development at 2-year intervals (2013–2018) revealed that the prevalence of HBV in patients with diabetes increased from 1.5% to 3.0% in two cycles (2013–2016), and the non-diabetic group also showed an increased tendency from 1.3% to 1.7%. However, compared with the 2013–2016 period, the HBV prevalence in 2017–2018 decreased to 2.4% in patients with diabetes and 1.1% in the non-diabetic group. The prevalence of HCV in the diabetic population decreased over time (2013–2018), though the non-diabetic group showed similar HCV development for two cycles (2013–2016) and indicated an increasing tendency towards 2.3% from 2017 to 2018. In the diabetic group, the hepatitis B or C prevalence increased from 5.0% to 5.7% during 2013–2016, while in 2017–2018, a decreasing trend in the rate was observed (Table 2 and Appendix A).

### 3.4. Risk of Hepatitis Development in Diabetes Patients 

In univariate and multivariate Cox regression analysis results, the non-poverty group showed a tendency to have a lower risk of hepatitis B or C prevalence than the poverty group (univariate: HR = 0.48; 95% CI, 0.31–0.74, *p* < 0.01; multivariate: HR = 0.50; 95% CI, 0.32–0.79, *p* < 0.01, respectively) (Table 3). In addition, diabetes patients who did not use illicit drugs had a lower risk of hepatitis B or C infection than illicit drug users (univariate: HR = 0.05; 95% CI, 0.03–0.08, *p* < 0.01; multivariate: HR = 0.05; 95% CI, 0.03–0.08, *p* < 0.01, respectively). In multivariate analysis, non-Hispanic Asian particularly showed a greater association in hepatitis B or C prevalence than other races among the diabetic group. (HR = 3.33; 95% CI, 1.05–10.60, *p* < 0.05).

### 3.5. Factors of Developing Hepatitis in Diabetes Patients

The results of the univariate and multivariate logistic regression analyses are summarized in Table 4. In the diabetic group, the univariate and multivariate logistic analyses showed that the non-poverty group was associated with a lower risk of hepatitis B or C development than the poverty group (univariate: OR 0.96; 95% CI, 0.94–0.99, *p* < 0.01; multivariate: OR 0.96; 95% CI, 0.94–0.99, *p* < 0.01, respectively). The diabetic group of non-users of illicit drugs was associated with reduced odds of hepatitis B or C infection compared with illicit drug users (univariate: OR 0.59; 95% CI, 0.55–0.62, *p* < 0.01; multivariate: OR 0.59; 95% CI, 0.56–0.62, *p* < 0.01, respectively). The non-Hispanic Asian group showed a tendency of higher correlation with hepatitis B or C infection than other races (univariate: OR 1.03; 95% CI, 0.97–1.09, *p* = 0.40; multivariate: OR 1.05; 95% CI, 0.99–1.11, *p* = 0.10, respectively).

## 4. Discussion

The current study evaluated the prevalence of HBV and HCV infections in patients with diabetes and relevant factors. Our results showed that patients with diabetes had a higher prevalence of HBV and HCV infections than those without. Similarly, a previous cohort study conducted in the United Kingdom observed a higher prevalence of HBV in patients with diabetes than in those without diabetes [27]. Another previous study from Pakistan Hospital also indicated that patients with diabetes were more associated with HCV infection than the non-diabetic group (OR = 3.03; 95% CI, 2.64–3.48, *p* < 0.01) [28]. Furthermore, due to their weakened immune system [29], patients with diabetes may be more susceptible to HBV or HCV infections compared to the general population. This susceptibility may be exacerbated by frequent exposure to needles during blood glucose monitoring, which can contribute to virus transmission [30].

Our findings demonstrated that poverty was a potential contributor that might influence HBV and HCV infection development. Patients with diabetes living in poverty were more associated with a higher development of hepatitis than those without poverty. Consistent with these findings, a previous cross-sectional study in Canada [31], which reported the association between socioeconomic income and the prevalence of diabetes and related conditions, also showed that those with a low household income had a higher prevalence of diabetes than the population with a higher income. Greene et al. [32] analyzed surveillance data in New York City and reported that chronic hepatitis C was included in diseases related to severe poverty with low income. Furthermore, based on the American Association for the Study of Liver Disease (AASLD), HBV and HCV are the leading infectious diseases that are closely related to poverty [33]. Although various factors contributed to poverty, such as age or education levels, and are closely related to infectious disease spread [34], considering the low self-awareness of their condition among impoverished people, the poverty group could show a progressive increase in the risk of disease such as hepatitis [35]. In support of this, in Brazil, a previous study also showed hepatitis B susceptibility rates ranging from approximately 32% among individuals living with low income [36]. Therefore, poverty in people with complications such as DM is the cause of increasing HBV or HCV and is also likely to be an important factor in the incidence rate because there is a cost burden [37]. 

In our results, patients with diabetes without illicit drugs seem to have a lower risk of hepatitis infection than illicit drug users. Diabetic patients who are at risk of experiencing high stress and immune dysfunction may also be more susceptible to illegal drug use and HCV transmission [38,39]. For the association between HCV transmission and illicit drug use, Benjamin et al. [40] already demonstrated in a study of young American users of illegal drugs that 343 out of 714 participants were infected with HCV. Furthermore, the increase in the distribution of hepatitis caused by illegal drug use seems to be a global burden [41]. This is evidenced by the population-attributable fraction of hepatitis caused by illegal drug use in 2013, which was 10% for HBV in North America and 1% in Latin America and 81% for HCV in North America and 31% in Latin America [41]. This percentage has further increased since 1990, with HBV accounting for 6% in North America and 1% in Latin America and HCV accounting for 60% in North America and 19% in Latin America [41]. However, given the tendency for illicit drug users to frequently disregard physician advice, hepatitis infections in this population may become more severe or even go undetected [42]. Therefore, these findings can offer crucial insights to enhance screening protocols and identify a broader population of illegal drug users at high risk of acquiring HCV and HBV infections. This will aid in the early detection and treatment of the infections [43].

Among races, non-Hispanic Asians race seemed to significantly contribute to hepatitis in the diabetic group. Consistent with these findings, a previous study showed that non-Hispanic Asians were more associated with the prevalence of HBV infection than other races (OR = 3.85; 95% CI, 2.97–4.97, *p* < 0.05) [44]. In addition, according to 2011–2014 NHANES data, non-Hispanic Asian adults showed a higher prevalence of HBV infection (22.6%) than non-Hispanic White (2.6%), non-Hispanic Black (10.2%), and Hispanic (3.6%) adults [45]. Furthermore, in our study, the non-Hispanic Black race seemed to contribute to hepatitis infection without significance. Pathologically, according to Thomas et al., interleukin 28 B (IL28B), which plays a significant primary role in the resolution of HCV, is less likely to be present in the Black population [46]. Because of the lack of IL28B, the non-Hispanic Black group may have a higher risk of HCV infection [46]. However, our study cannot be confidential because race affects the risk of hepatitis; therefore, further studies are needed.

The current study had some limitations. First, our study did not distinguish between type 1 and type 2 diabetes mellitus. As type 2 diabetes accounts for more than 90% of diagnosed diabetes mellitus in the United States [47], our findings largely reflect the risk factors of hepatitis in patients with type 2 diabetes mellitus. Thus, we expect that more future studies will be conducted to distinguish the types of diabetes mellitus to identify the impact of diabetes and hepatitis. Second, we did not evaluate the economic costs of treating HCV infection and diabetes mellitus. Therefore, further studies are required to evaluate the economic impact of HBV or HCV infection in patients with diabetes. Third, we were unable to conduct further analysis on the correlation between hepatitis infection and CVD or diabetic comorbidities in the current study. This was because the important cardiovascular health metrics data, which could aid analysis of the correlation of hepatitis infection with complications such as cardiovascular disease or diabetic complications, were scarce in the NHANES [48], or the guidelines for evaluating criteria for high blood pressure and cholesterol risks were changed [49,50]. Thus, we hope that future studies can address these limitations. Forth, the NHANES data were limited to the United States. This may not provide global evidence for an association between diabetes and hepatitis. Thus, we expect further studies using merged data from various countries. Finally, as this study used a small sample size to identify illegal injection drug users, the number might have been underestimated [51]. It is difficult to obtain accurate data about illegal drug users. Thus, there may be inevitable non-responsive biases with illicit drug use, and our findings should be interpreted with caution.

## 5. Conclusions

The current study found that the risk of hepatitis B or C infection was higher in patients with diabetes than in those without. Therefore, the present study would help increase awareness regarding hepatitis prevention in patients with diabetes. Additionally, this study suggests that more attention should be paid to impoverished or illicit drug users among patients with diabetes regarding the threat of hepatitis infection. Finally, in addition to NHANES data, we expect that more global evidence will be provided through corresponding data from various countries.

## Figures and Tables

**Figure 1 ijerph-20-04962-f001:**
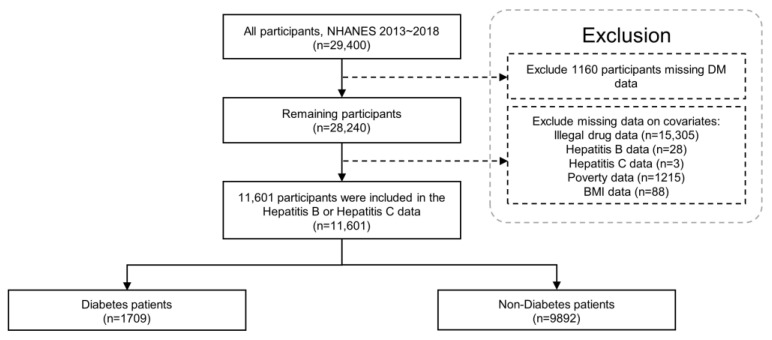
Flowchart of the study participants using the NHANES 2013–2018. BMI: body mass index; NHANES: National Health and Nutrition Examination Survey.

**Figure 2 ijerph-20-04962-f002:**
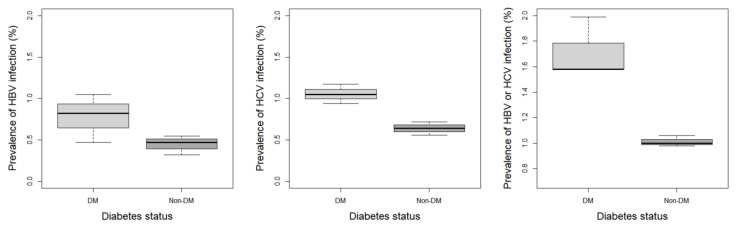
Prevalence of each hepatitis regarding the diabetes mellitus status. DM, diabetes mellitus; HBV, hepatitis B virus; HCV, hepatitis C virus.

**Table 1 ijerph-20-04962-t001:** Baseline characteristics of the study population from the National Health and Nutrition Examination Survey (NHANES) 2013–2018.

Variables	2013–2014	2015–2016	2017–2018	All (2013–2018)
Age (n)				
0~19	286	207	203	696
20~44	1999	1849	1573	5421
45~64	1556	1449	1501	4506
≥65	337	317	324	978
Sex (n)				
Female	2158	1966	1862	5986
Male	2020	1856	1739	5615
BMI (n)				
Underweight (<18.5 kg/m²)	71	64	64	199
Normal (18.5~24.9 kg/m²)	1272	1035	913	3220
Obese (≥25 kg/m²)	2835	2723	2624	8182
Race (n)				
Mexican-American	605	714	520	1839
Other Hispanic	384	501	328	1213
Non-Hispanic White	1709	1208	1197	4114
Non-Hispanic Black	876	829	845	2550
Non-Hispanic Asian	458	414	496	1368
Other Race—Including Multi-Racial	146	156	215	517
Poverty (n)				
Yes	988	863	736	2587
No	3190	2959	2865	9014
Hepatitis B (n)				
Yes	54	72	46	172
No	4124	3750	3555	11,429
Hepatitis C (n)				
Yes	83	73	87	243
No	4095	3749	3514	11,358
Hepatitis B or C (n)				
Yes	132	133	124	389
No	4046	3689	3477	11,212
Illicit Drug Use (n)				
Yes	103	82	100	285
No	4075	3740	3501	11,316

BMI: body mass index.

**Table 2 ijerph-20-04962-t002:** Characteristics of hepatitis infection status in with and without diabetes mellitus.

Variables	2013–2014						2015–2016						2017–2018						All (2013–2018)					
	With DM (n = 540), N (%)	Without DM (n = 3638), N (%)	OR	95% CI		*p*	With DM (n = 593), N (%)	Without DM (n = 3229), N (%)	OR	95% CI		*p*	With DM (n = 576), N (%)	Without DM (n = 3025), N (%)	OR	95% CI		*p*	With DM (n = 1709), N (%)	Without DM (n = 9892), N (%)	OR	95% CI		*p*
				Lower	Upper					Lower	Upper					Lower	Upper					Lower	Upper	
Hepatitis B (n, %)						0.677						0.025						0.007						0.002
Yes	8 (1.5%)	46 (1.3%)	1.174	0.551	2.502	0.677	18 (3.0%)	54 (1.7%)	1.841	1.072	3.161	0.025	14 (2.4%)	32 (1.1%)	2.330	1.235	4.394	0.007	40 (2.3%)	132 (1.3%)	1.772	1.239	2.534	0.002
No	532 (98.5%)	3592 (98.7%)	0.852	0.400	1.814	0.677	575 (97.0%)	3175 (98.3%)	0.543	0.316	0.933	0.025	562 (97.6%)	2993 (98.9%)	0.429	0.228	0.810	0.007	1669 (97.7%)	9760 (98.7%)	0.564	0.395	0.807	0.002
Hepatitis C (n, %)						0.002						0.029						0.537						0.001
Yes	20 (3.7%)	63 (1.7%)	2.183	1.309	3.640	0.002	18 (3.0%)	55 (1.7%)	1.807	1.053	3.099	0.029	16 (2.8%)	71 (2.3%)	1.189	0.686	2.060	0.537	54 (3.2%)	189 (1.9%)	1.675	1.232	2.277	0.001
No	520 (96.3%)	3575 (98.3%)	0.458	0.275	0.764	0.002	575 (97.0%)	3174 (98.3%)	0.554	0.323	0.950	0.029	560 (97.6%)	2954 (97.7%)	0.841	0.485	1.458	0.537	1655 (96.8%)	9703 (98.1%)	0.597	0.439	0.811	0.001
Hepatitis B or C (n, %)						0.009						0.001						0.074						<0.0001
Yes	27 (5.0%)	105 (2.9%)	1.771	1.149	2.730	0.009	34 (5.7%)	99 (3.1%)	1.923	1.289	2.868	0.001	27 (4.7%)	97 (3.2%)	1.485	0.960	2.296	0.074	88 (5.1%)	301 (3.0%)	1.730	1.356	2.206	<0.0001
No	513 (95.0%)	3533 (97.1%)	0.565	0.366	0.871	0.009	559 (94.3%)	3130 (96.9%)	0.520	0.349	0.776	0.001	549 (95.3%)	2928 (96.8%)	0.674	0.436	1.042	0.074	1621 (94.9%)	9591 (97.0%)	0.578	0.453	0.737	<0.0001

CI: confidence interval; DM: diabetes mellitus; OR: odds ratio; *p*: *p*-value.

**Table 3 ijerph-20-04962-t003:** Univariate and multivariate hazard ratio and 95% confidence intervals for the risk of hepatitis B or C with diabetes mellitus population.

Variables	Participants (n = 1709), N (%)	Univariate Analysis				Multivariate Analysis			
		HR	95% CI		*p*	HR	95% CI		*p*
			Lower	Upper			Lower	Upper	
Age									
0~19	9 (0.5%)	REF				REF			
20~44	308 (18.0%)	655530	0.000	INF	0.994	787400	0.000	INF	0.995
45~64	1057 (61.9%)	1315795	0.000	INF	0.993	1603000	0.000	INF	0.994
≥65	335 (19.6%)	1512302	0.000	INF	0.993	2181000	0.000	INF	0.994
Race									
Mexican-American	343 (20.1%)	REF				REF			
Other Hispanic	199 (11.6%)	1.483	0.499	4.414	0.479	1.356	0.454	4.051	0.586
Non-Hispanic White	477 (27.9%)	2.875	1.252	6.603	0.013	2.578	1.113	5.972	0.027
Non-Hispanic Black	447 (26.2%)	3.128	1.373	7.122	0.007	2.815	1.229	6.446	0.014
Non-Hispanic Asian	171 (10.0%)	3.540	1.427	8.781	0.006	5.310	2.060	13.178	0.000
Other Race– Including Multi-Racial	72 (4.2%)	2.230	0.652	7.624	0.201	1.563	0.449	5.440	0.483
Poverty									
Yes	1296 (75.8%)	REF				REF			
No	413 (24.2%)	0.477	0.309	0.735	0.001	0.500	0.320	0.789	0.003
Illicit Drug Use									
Yes	1660 (97.1%)	REF				REF			
No	49 (2.9%)	0.050	0.032	0.079	<0.0001	0.050	0.031	0.080	<0.0001

CI: confidence interval; HR: hazard ratio; INF: infinite; *p*: *p*-value; REF: reference.

**Table 4 ijerph-20-04962-t004:** Univariate and multivariate odds ratio and 95% confidence intervals for the association of hepatitis B or C with the diabetes mellitus population.

Variables	Participants (n = 1709), N (%)	Univariate Analysis				Multivariate Analysis			
		OR	95% CI		*p*	OR	95% CI		*p*
			Lower	Upper			Lower	Upper	
Age									
0~19	9 (0.5%)	REF				REF			
20~44	308 (18.0%)	1.026	0.887	1.188	0.728	1.023	0.895	1.169	0.742
45~64	1057 (61.8%)	1.057	0.915	1.222	0.450	1.053	0.923	1.202	0.444
≥65	335 (19.6%)	1.065	0.920	1.232	0.401	1.067	0.934	1.220	0.338
Race									
Mexican-American	343 (20.1%)	REF				REF			
Other Hispanic	199 (11.6%)	1.010	0.972	1.049	0.620	1.008	0.973	1.044	0.672
Non-Hispanic White	477 (27.9%)	1.037	1.006	1.069	0.021	1.029	1.001	1.059	0.042
Non-Hispanic Black	447 (26.2%)	1.048	1.016	1.081	0.003	1.042	1.013	1.072	0.005
Non-Hispanic Asian	171 (10.0%)	1.063	1.021	1.107	0.003	1.079	1.040	1.120	<0.0001
Other Race— Including Multi-Racial	72 (4.2%)	1.036	0.979	1.095	0.219	1.030	0.979	1.084	0.259
Poverty									
Yes	1296 (75.8%)	REF				REF			
No	413 (24.2%)	0.963	0.940	0.987	0.003	0.963	0.941	0.985	0.001
Illicit Drug Use									
Yes	1660 (97.1%)	REF				REF			
No	49 (2.9%)	0.586	0.553	0.620	<0.0001	0.588	0.555	0.623	<0.0001

CI: confidence interval; OR: odds ratio; *p*: *p*-value; REF: reference.

## Data Availability

The public-use NHANES data analyzed in this study are available at https://www.cdc.gov/nchs/nhanes/index.htm accessed on 10 September 2022.

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
