# Peer review of "Hepatitis Risk in Diabetes Compared to Non-Diabetes and Relevant Factors: A Cross-Sectional Study with National Health and Nutrition Examination Survey (NHANES), 2013–2018"

_ijerph, 2023, doi:10.3390/ijerph20064962_

Round 1

Reviewer 1 Report

The present study investigated the development of hepatitis B or C infection in diabetes patients compared to those without and to elucidate factors associated with the prevalence of hepatitis B or C infection in diabetes. The results showed that in patients with diabetes, the development of hepatitis was higher than that in those without, and hepatitis development was influenced by poverty and illicit drug use. The study is of clinical significance. Some minor points are listed as below.

1. Now that the prevalence of hepatitis was differed regarding the diabetes mellitus status, was the prevalence correlated the fast glucose or glycated hemoglobin levels?

2. Were the prevalence of hepatitis influenced by comorbidities such as cardiovascular diseases and diabetic complications?

Reviewer 2 Report

This study uses the NHANES dataset to assess the prevalence of hepatitis (B and C) among people with diabetes. People with multiple conditions would be important to target for interventions, and so this information is valuable, however I have some notes that the authors should clarify. In particular it wasn’t entirely clear what types of data was being used and how some calculations were conducted.

Major comments

1.      It should be clarified more about the setting of the analysis, it is not clear that this study is about the US

2.      It is not clear whether the NHANES data used presents serial cross-sectional information, or a longitudinal cohort. Both would make the interpretation of the results significantly different. If it is serial cross sectional, then the study should not be calculating or reporting on incidence, or hazard ratios (i.e. the data might just be changes in prevalence of the surveys over time).

3.      The exclusion of more than half of the dataset, due to missing illicit drug use data is slightly concerning. How is this variable collected, and is there a bias if only some people are asked about this? Were the variables similar for people with vs without responses for illicit drug use variable missing?

4.      In the discussion, there is a lot of conjecture about causal relationships between different socioeconomic factors and development of both hepatitis and diabetes. For example, it seems unlikely that increased hospital visits would result in increased risk of HBV/HCV infection, given there are likely to be safe injecting practices in place in these facilities? Also, some of these things could go the other way. For example, people who inject drugs may live in poverty, and poverty may limit access to healthcare and hence diabetes may be a consequence. A suggestion would be to consider revising the discussion to avoid drawing any causal conclusions, or to at least confine conjecture on such relationships to a paragraph that clearly states it is a discussion of unknown pathways. Perhaps the discussion could include focus on the implications of the results and what they might mean for future treatment or testing targeting?

Introduction

5.      Is the 17% among patients with diabetes and viral hepatitis compared to patients with diabetes only, or compared to patients with neither?

6.      “865 outbreaks of hepatitis B” – how is an outbreak defined, and in what geographical context (e.g. globally, regionally, country-level)?

7.      Lines 39-41: it is unclear why patients with diabetes are more vulnerable to HCV infection that those without diabetes, is this suggesting that people with diabetes inject drugs more, and is there data to support that?

8.      Lines 43-44: should this be the risk of acquiring HBV or HCV if exposed? Since the risk of HBV and HCV will be dependent on behaviours rather than biology

Methods

9.      It is true that the NHANES details are on the website, but it would be helpful none the less to have some minimal details provided

10.   Line 94: I think this should be >=1 rather than >=117

11.   Statistical analyses: why were only some variables included?

12.   Why were separate models chosen for people with vs without diabetes, and not a single model with diabetes as an independent variable?

Results

13.   Do the data include the same people in multiple years?

14.   Section 3.2: I don’t think it is correct to say “X times more” when it is an odds ratio. Suggest changing working to increased odds or similar.

15.   Figure 2: what are the boxes showing the distribution of, is this across years?

16.   Section 3.2: how is “incidence rate” calculated, or are the authors referring to prevalence?

17.   Table 3: is there a typo with the age categories?

Round 2

Reviewer 2 Report

Thank you for considering my previous comments